# InNeRF: Learning Interpretable Radiance Fields for Generalizable 3D Scene Representation and Rendering

## ABSTRACT

We propose Interpretable Neural Radiance Fields (InNeRF) for generalizable 3D scene representation and rendering. In contrast to previous image-based rendering, which used two independent working processes of pooling-based fusion and MLP-based rendering, our framework unifies source-view fusion and target-view rendering processes via an end-to-end interpretable Transformer-based network. InNeRF enables the investigation of deep relationships between the target-rendering view and source views that were previously neglected by pooling-based fusion and fragmented rendering procedures. As a result, InNeRF improves model interpretability by enhancing the shape and appearance consistency of a 3D scene in both the surrounding view space and the ray-cast space. For a query rendering 3D point, InNeRF integrates both its projected 2D pixels from the surrounding source views and its adjacent 3D points along the query ray and simultaneously decodes this information into the query 3D point representation. Experiments show that InNeRF outperforms state-of-the-art image-based neural rendering methods in both scene-agnostic and per-scene finetuning scenarios, especially when there is a considerable disparity between source views and rendering views. The interpretation experiment shows that InNeRF can explain a query rendering process.

## CCS CONCEPTS

• **Computing methodologies** → **Computer vision**; **Rendering**.

## KEYWORDS

Neural Rendering, Network Interpretability

## 1 INTRODUCTION

Novel view synthesis is a long-standing open problem concerned with the rendering of unseen views of a 3D scene given a set of observed views [16, 21]. Recent remarkable NeRF research [11, 12, 14, 18, 30] introduces neural radiance field scene representations, which use multi-layer perceptrons (MLPs) to map a continuous 3D location and view direction to its density and color.

However, these models need to optimize a specific 3D representation for each scene, which is time-consuming and does not learn the shared information among scenes. Subsequently, to learn prior knowledge in diverse scenes, researchers [4, 22, 25, 29] generalize the radiance field scene representation by incorporating a

*ACM MM, 2024, Melbourne, Australia*
© 2024 Copyright held by the owner/author(s). Publication rights licensed to ACM.
ACM ISBN 978-x-xxxx-xxxx-x/YY/MM
https://doi.org/10.1145/nnnnnnn.nnnnnnn

pooling-based multi-view feature as the conditional input. These prior NeRFs generally contain three basic components: a CNN-based single-view feature extraction module, a pooling-based multi-view fusion module, and an MLP-based NeRF module.

Despite the intrinsic connection between these modules, each module is designed and studied independently, making the overall framework disjointed. This incoherent framework design damages the model interpretability from three aspects: 1) Separating feature extraction of each source view overlooks their relevancy in representing 3D scenes. 2) Pooling-based fusion cannot fully explore the complicated relationship among source views. 3) The MLP network rendering the color and density from a single aggregated feature struggles to decode intricate relationships between observed views and the rendering view. The reason for this framework design is that previous NeRFs are built on MLPs that are incapable of processing an arbitrary number of observed views. Consequently, they need an auxiliary fusion model to aggregate multi-view information, and pooling-based fusion provides such a straightforward technique.

This limitation also impairs the capability of NeRFs to learn a view-consistent 3D scene representation from observed views, especially for the scenario where source views have a more complicated relationship with the target view, e.g. the observed source views are captured at camera poses that are very different from the camera pose of the target view. When camera poses of source views are similar to the rendering view, source views and the target view are distributed in a local region in 3D scene representation space, making it possible to approximate their relationship by a linear function as in previous work [4, 22, 25, 29]. However, as the difference between observed views and the rendering view increases, the correlation becomes more complicated, making it challenging for these approaches to synthesize a realistic novel view. In this scenario, existing MLP-based NeRFs, using a pooling-based function to fuse the multi-view, are insufficient to tackle this challenge.

Therefore, the fundamental issue is how to free the intrinsic interpretability of NeRFs from the previously fragmented frameworks for learning generalizable radiance fields. To tackle this unmet need, we present Interpretable Neural Radiance Fields (InNeRF), an end-to-end Transformer-based architecture that unifies source-view fusion and target-view rendering processes for generalizable 3D scene representation and rendering. In the rendering process of a query 3D point, InNeRF is divided into two stages: the first works in the surrounding-view space, integrating information of the projected 2D pixels at the surrounding source views for the query 3D point; and the second works in the ray-cast space, fusing the neighboring 3D points along the query ray into the representation of the query 3D point, as shown in Fig 1. This design provides our model with a comprehensive understanding of the shape and appearance consistency of a 3D scene in both the surrounding-view space and ray-cast spaces. Furthermore, the Transformer-based framework taking advantage of the attention mechanism enables our rendering

process to learn in-depth and complicated relationships between source views and the rendering view, which is essential for novel view synthesis. Therefore, InNeRF has improved interpretability and learns a more comprehensive general neural radiance field.

Our contributions can be summarized as follows:

- We propose Interpretable Neural Radiance Fields (InNeRF), a unified Transformer-based framework, to study deep correlations between observed and rendering views and simultaneously integrate this intricate information into a generalizable neural radiance field.
- InNeRF exploits geometry and appearance consistency of a neural radiance field in both the surrounding view space and the ray-cast space, strengthening its interpretability.
- Experiments show that InNeRF achieves more realistic rendering results than state-of-the-art methods in both scene-agnostic and per-scene fine-tuning settings, especially when source views are captured at camera poses that differ significantly from the rendering view.
- InNeRF explains a query rendering process by utilizing its attention layers. Experiments show that the interpretation of InNeRF is consistent with human perception.

## 2 RELATED WORK

**Novel View Synthesis.** The goal of novel view synthesis is to render unseen views of a scene from its multiple observed images. The essence of novel view synthesis is exploring and learning a view-consistent 3D scene representation from a sparse set of input views. The early work focused on modeling 3D shapes by discrete geometric 3D representations, such as mesh surface [7, 8, 17], point cloud [10, 19] and voxel grid [1, 24, 28]. Although explicit 3D geometry-based representations are intuitive, they are discrete and sparse, making them incapable of learning high-resolution renderings with sufficient quality for complex scenes.

More recently, the impressive neural radiance field (NeRF) [16] has shown a solid ability to synthesize novel views by representing continuous scenes as 5D radiance fields in MLPs. Nevertheless, NeRF optimizes each scene representation independently, not exploring the shared information amongst scenes and being time-consuming. Subsequently, researchers proposed models, such as PixelNeRF [29], MVSNeRF [4], IBRNet [25], which receives as conditional inputs multiple observed views to learn a general neural radiance field. These methods are proposed using the divide-and-conquer strategy and have two separate components: a CNN feature extractor for each observed image and an MLP as the NeRF network. However, pooling-based fusion models in these methods barely explore the complex relationship across multiple views for 3D scene understanding. Furthermore, processing each 3D point independently ignores the geometry consistency of a 5D radiance field of a scene.

Here, we propose an encoder-decoder Transformer framework, InNeRF, to represent the neural radiance field scene for novel view synthesis. Compared with the pooling-based fusion in previous work, InNeRF can explore deep relationships among multiple views and aggregate multi-view information into the coordinate-based scene representation by the attention mechanism in a unified network. Meanwhile, InNeRF can learn the consistency of shape and appearance in a scene by considering the corresponding information in the surrounding view space and the ray-cast space.

**Transformer.** Transformer recently emerged as a promising network framework and has achieved impressive performance in natural language processing [2, 20, 27] and computer vision [3, 5, 6, 9, 13, 31]. The main idea behind this approach is to utilize the multi-head self-attention operation to explore the dependence within input tokens and learn a global feature representation. In the object detection task, DETR [3] presents a new framework combining a 2D CNN with a Transformer and predicts object detection in parallel as a sequence of output tokens. In image classification, ViT [6] demonstrates the impressive ability to learn global contexts in Transformer even without using CNN features [23]. In 3D scene understanding, FlatFormer [13] introduces a new window attention mechanism to optimize the computational efficiency and achieve improved performance in reconstruction.

For novel view synthesis, we introduce an end-to-end Transformer framework to implicitly model the continuous 3D scene as a neural radiance field representation. Our model leverages the advantage of Transformer in exploring deep relationships among observed images to learn a consistent generalizable 3D scene representation.

## 3 METHODOLOGY

### 3.1 Framework

We propose InNeRF to learn an interpretable generic radiance field representation for novel scenes. Given captured multi-view images $\{\mathbf{I}^m\}_{m=1}^M$ ($M$ source views) of diverse scenes and their camera parameters $\{\Theta^m\}_{m=1}^M$ (camera poses, intrinsic parameters and scene bounds), InNeRF reconstructs a generic radiance field $F_{\text{InNeRF}}$ to learn the prior knowledge:

$$(\sigma, \mathbf{c}) \leftarrow F_{\text{InNeRF}}((x, y, z), \mathbf{d}; \{\mathbf{I}^m, \Theta^m\}_m) , \tag{1}$$

where $(x, y, z)$ is a 3D point location, $\mathbf{d}$ denotes a unit-length direction of a viewing ray and outputs are a differential volumetric density $\sigma$ and a directional emitted color $\mathbf{c}$.

As shown in Fig. 1, for rendering a query 3D point on a target-viewing ray, the proposed InNeRF includes two stages: 1) In the surrounding-view space, our $\text{Decoder}_\sigma^{views}$ (Sec. 3.2) and $\text{Decoder}_c^{views}$ (Sec. 3.4) fuse source views and query spatial information ($(x, y, z)$, $\mathbf{d}$) into the latent density and color representations for the query point; 2) In the ray-cast space, we use $\text{Decoder}_\sigma^{ray}$ (Sec. 3.3) and $\text{Decoder}_c^{ray}$ (Sec. 3.5) to enhance the query density and color representations by considering neighboring points along the target ray. Finally, we obtain the density and color for the query point on a target-viewing ray.

### 3.2 Density Decoder in Surrounding-view Space

We first present our density decoder in surrounding-view space ($\text{Decoder}_\sigma^{views}$) decoding the projected pixels at source views into the query latent density code.

For each source view, we first extract its feature volume by a pre-trained view-shared U-Net. A query 3D point $(x, y, z)$ is then projected into each source view $\mathbf{I}^m$ by its camera projection matrix $\Theta^m$ to extract the corresponding RGB color $\{\mathbf{c}_{src}^m\}_{m=1}^M$ and feature vector $\{\mathbf{f}_{src}^m\}_{m=1}^M$ at the projected 2D pixel $\{\mathbf{p}^m\}_{m=1}^M$ location

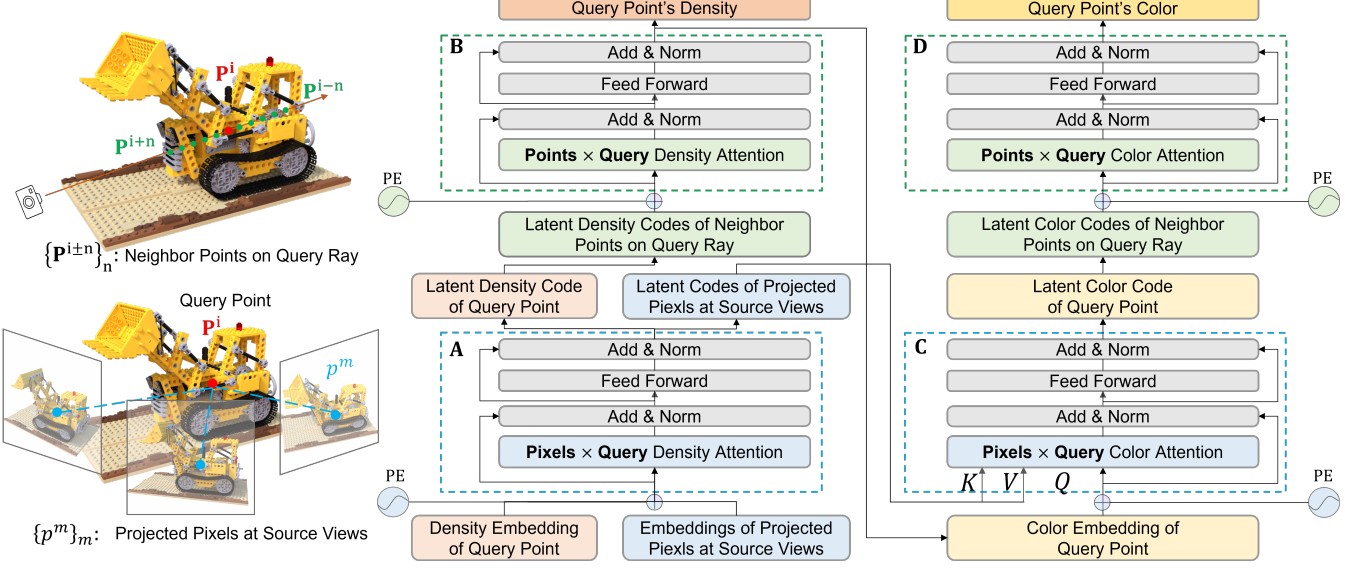

**Figure 1: Workflow of the proposed InNeRF. Module A is the density decoder in surrounding-view space (Sec. 3.2). Module B is the density decoder in ray-cast space (Sec. 3.3). Module C is the color decoder in surrounding-view space (Sec. 3.4). Module D is the color decoder in ray-cast space (Sec. 3.5).**

through bilinear interpolation. In each source view, we also record its viewing direction $\{\mathbf{d}_{src}^m\}_{m=1}^M$ for the projected pixel from the source camera pose. Based on it, we obtain the initial source-view embeddings $\{\mathbf{x}_0^m\}_{m=1}^M$ for source views.

For the query point, $\text{Decoder}_\sigma^{views}$ receives the initial source-view embeddings $\{\mathbf{x}_0^m\}_{m=1}^M$ and the learnable query density embedding $\mathbf{x}_0^\sigma$ as inputs $\mathbf{X}_0$. $\text{Decoder}_\sigma^{views}$ can be formulated as follows:

$$\mathbf{X}_0 = [\mathbf{x}_0^\sigma; \mathbf{x}_0^1; \mathbf{x}_0^2; \cdots ; \mathbf{x}_0^M] , \tag{2}$$

$$\tilde{\mathbf{X}}_{l+1} = \text{Norm}(\text{Pixels} \times \text{Query}_\sigma(\mathbf{X}_l) + \mathbf{X}_l) , \tag{3}$$

$$\mathbf{X}_{l+1} = \text{Norm}(\text{FFN}(\tilde{\mathbf{X}}_{l+1}) + \tilde{\mathbf{X}}_{l+1}) , \tag{4}$$

where $l$ denotes the index of a basic block ($l = 1, \cdots, L$), "Norm" is a layer normalization function, and "FFN" is a position-wise feed-forward network. At the $L$-th block, we can obtain $\mathbf{X}_L = [\mathbf{x}_L^\sigma; \mathbf{x}_L^1; \mathbf{x}_L^2; \cdots ; \mathbf{x}_L^M]$. In $\text{Decoder}_\sigma^{views}$, we concatenate the embedding $\mathbf{x}_L^\sigma$ and its 3D coordinate location $(x, y, z)$ as the latent density code for the query point.

Pixels×Query Density Attention layers explore deep relationships among source views, defined as follows:

$$\text{Pixels} \times \text{Query}_\sigma(\mathbf{X}) = \text{MH-Attn}(\mathbf{X}, \mathbf{X}, \mathbf{X}) , \tag{5}$$

where the multi-head attention function is defined as:

$$\text{MH-Attn}(\mathbf{Q}, \mathbf{K}, \mathbf{V}) = \text{Cat}(\mathbf{A}_1, \cdots, \mathbf{A}_H)\mathbf{W} , \tag{6}$$

$$\text{where } \mathbf{A}_h = \text{Attention}(\mathbf{Q}_h, \mathbf{K}_h, \mathbf{V}_h) ,$$

$$\mathbf{Q}_h = \mathbf{Q}\mathbf{W}_h^q; \mathbf{K}_h = \mathbf{K}\mathbf{W}_h^k; \mathbf{V}_h = \mathbf{V}\mathbf{W}_h^v .$$

Here, $\mathbf{W}_h^q, \mathbf{W}_h^k \in \mathbb{R}^{d_k \times d_h}; \mathbf{W}_h^v \in \mathbb{R}^{d_v \times d_h}$ and $\mathbf{W} \in \mathbb{R}^{Hd_h \times d_k}$ are parameter matrices ($H \times d_h = d_k$ and $d_h$ is the feature dimension

in each head). The Attention function is computed by

$$\text{Attention}(\mathbf{Q}, \mathbf{K}, \mathbf{V}) = \text{softmax}(\frac{\mathbf{Q}\mathbf{K}^T}{\sqrt{d_k}})\mathbf{V} , \tag{7}$$

Here, $N_q$ queries are stacked in $\mathbf{Q} = [\mathbf{q}_1; \mathbf{q}_2; \cdots ; \mathbf{q}_{N_q}] \in \mathbb{R}^{N_q \times d_k}$, a set of $N_k$ key-value pairs are stacked in $\mathbf{K} = [\mathbf{k}_1; \mathbf{k}_2; \cdots ; ; \mathbf{k}_{N_k}] \in \mathbb{R}^{N_k \times d_k}$ and $\mathbf{V} = [\mathbf{v}_1; \mathbf{v}_2; \cdots ; \mathbf{v}_{N_k}] \in \mathbb{R}^{N_k \times d_v}$, $d_k$ is used as a scalar for normalization. Our $\text{Decoder}_\sigma^{views}$ is invariant to permutations of source views and can receive an arbitrary number of source views.

## 3.3 Density Decoder in Ray-cast Space

The density decoder in ray-cast space ($\text{Decoder}_\sigma^{ray}$) decodes the density information of the query point by aggregating the density features of the neighboring 3D points along the target-view ray.

For the query point and neighboring $2n$ points along the target-viewing ray, we denote $[\sigma_0^{i-n}; \cdots \sigma_0^i \cdots ; \sigma_0^{i+n}]$ as their initial density representations at the input end of the $\text{Decoder}_\sigma^{ray}$, where the query point is denoted as $P^i$ and neighboring $2n$ points are $\{P^{i-n}, \cdots, P^{i-1}, P^{i+1}, \cdots, P^{i+n}\}$. Here, the initial density representation for each 3D point is computed via an FC layer based on the $\text{Decoder}_\sigma^{views}$ output for the corresponding point ($\sigma_0 = \text{FC}(\mathbf{x}_L^\sigma \odot (x, y, z))$, where $\odot$ is the concatenation operation). Then positional encodings $\mathbf{E}^{pos}$ are added to density representations of neighboring points to keep their position information in the ray-cast space. Each positional encoding informs each point of its 3D spatial location, which is computed by utilizing sine and cosine functions of different frequencies as [3].

Decoder$_\sigma^{ray}$ is formulated as follows:

$$D_0 = [\sigma_0^{i-n}; \cdots \sigma_0^{i} \cdots ; \sigma_0^{i+n}] + E^{pos} , \qquad (8)$$

$$\tilde{D}_{l+1} = \text{Norm}(\text{Points} \times \text{Query}_\sigma(D_l) + D_l) , \qquad (9)$$

$$D_{l+1} = \text{Norm}(\text{FFN}(\tilde{D}_{l+1}) + \tilde{D}_{l+1}) , \qquad (10)$$

where the Points×Query Density Attention layer is computed as Points×Query$_\sigma$ = MH-Attn$(D, D, D)$ fusing information of surrounding 3D points on the target-viewing ray. At the end block, the Decoder$_\sigma^{ray}$ outputs the density representation $\sigma_L^i$ of the query 3D point, and then we use an FC layer to project it to the density value.

### 3.4 Color Decoder in Surrounding-view Space

The color decoder in surrounding-view space (Decoder$_c^{views}$) decodes the projected pixels' information from source views into the query color representation.

Decoder$_c^{views}$ can be formulated as follows:

$$\tilde{Y}_{l+1} = \text{Norm}(\text{Pixels} \times \text{Query}_c(Y_l, \hat{X}, \hat{C}) + Y_l) , \qquad (11)$$

$$Y_{l+1} = \text{Norm}(\text{FFN}(\tilde{Y}_{l+1}) + \tilde{Y}_{l+1}) . \qquad (12)$$

In Pixels×Query Color Attention layers, the initial query color embedding is represented as $Y_0 = \text{FC}(\sigma_L^i) \odot d_{tgt}$, where $\sigma_L^i$ is the latent density representation from Decoder$_\sigma^{ray}$ and $d_{tgt}$ is the target-viewing direction for the query point. Pixels×Query Color Attention layer is calculated as:

$$\text{Pixels} \times \text{Query}_c(Y, \hat{X}, \hat{C}) = \text{MH-Attn}(Y, \hat{X}, \hat{C}) , \qquad (13)$$

where the value is $\hat{C} = [\gamma(c_{src}^1); \cdots ; \gamma(c_{src}^M)]$ ($\gamma(\cdot)$ is the embedding function) and the key is $\hat{X} = [\text{FC}(x_L^1) \odot d_{src}^1; \cdots ; \text{FC}(x_L^M) \odot d_{src}^M]$ representing the projected pixels' representations in source views. The output $Y_L$ is the latent color code for the query 3D point.

### 3.5 Color Decoder in Ray-cast Space

The color decoder in ray-cast space (Decoder$_c^{ray}$) learns a query color by fusing latent color codes of adjacent 3D points along the target ray in Points×Query Color Attention layers (Points×Query$_c(Z)$ = MH-Attn$(Z, Z, Z)$). Decoder$_c^{ray}$ is represented as:

$$Z_0 = [z_0^{i-n}; \cdots z_0^{i} \cdots ; z_0^{i+n}] + E^{pos} , \qquad (14)$$

$$\tilde{Z}_{l+1} = \text{Norm}(\text{Points} \times \text{Query}_c(Z_l) + Z_l) , \qquad (15)$$

$$Z_{l+1} = \text{Norm}(\text{FFN}(\tilde{Z}_{l+1}) + \tilde{Z}_{l+1}) . \qquad (16)$$

where the query latent color code from Decoder$_c^{views}$ is assigned to the corresponding $z_0^i$ and likewise for adjacent $2n$ points in ray-cast space.

Subsequently, after the Decoder$_c^{ray}$, we use an FC layer to project the output color embedding $z_L^i$ to its output predicted color value. Then the predicted density and color of each query point along a ray of the desired virtual camera are put forward to the classical volume rendering. The implementation details of the network and training are described in the supplementary material.

## 4 EXPERIMENTS

The proposed approach is evaluated in the following experimental settings:

- Scene-agnostic setting: we train a single scene-agnostic model on a large training dataset that includes various camera setups and scene types. We test its generalization ability to unseen scenes on all test scenes.
- Per-scene fine-tuning setting: our pretrained scene-agnostic model is finetuned on each test scene. We evaluate each finetuned scene-specific model separately.

We train and evaluate our method on a collection of multi-view datasets containing both synthetic data and real data, as in IBRNet [25]. For novel view synthesis, we quantitatively evaluate the rendered image quality based on PSNR, SSIM [26] (higher is better), and LPIPS [32] (lower is better).

### 4.1 Conditional Source-view Set

Experiments are designed to examine whether the proposed InNeRF can effectively learn a neural radiance field scene representation in scenarios where the variation degree between the conditional source view set and the target rendering view changes. Here, we sample 10 views from the surrounding view set as the conditional source-view set to render a target view. Given the camera pose, we can compute and sort the difference between each surrounding view and the target rendering view.

Based on the sorted order, we construct $N_s$ conditional source-view sets ($\{S_i\}_{i=1}^{N_s}$) from the surrounding-view set to render each test view. For the real evaluation dataset, there are $N_s$ = 3 sets, i.e. top 10 ($S_1$), middle 10 ($S_2$), and bottom 10 ($S_3$) views. For the synthetic evaluation dataset, there are $N_s$ = 4 sets which are the top 10 ($S_1$), middle 10 ($S_2$), 3/4th 10 ($S_3$), and bottom 10 ($S_4$) views, respectively. Fig. 4 shows visual examples of $S_1$ and $S_4$ for illustration.

### 4.2 Results

In both the scene-agnostic (Sec. 4.2.1) and per-scene fine-tuning experiments (Sec. 4.2.2), we evaluate competing methods in scenarios where the source views belong to different source view sets $\{S_i\}_{i=1}^{N_s}$ defined in Sec. 4.1. To render a testing view, each competing approach receives as input the same source-view set. In Sec. 4.2.3, we provide the interpretation results of InNeRF.

*4.2.1 Scene-agnostic Experiments.* In scene-agnostic experiments, InNeRF is compared with PixelNeRF [29], MVSNeRF [4] and IBRNet [25] on the real forward-facing dataset [15] and the realistic synthetic dataset [25].

Tab. 1 shows that the proposed InNeRF outperforms other methods on both datasets under the scene-agnostic setting. To facilitate the quantitative comparison in each metric, the best scores are marked in bold. It shows that InNeRF has a better generalization ability to novel scenes though it is trained on datasets with noticeably different scenes and view distribution. The detailed results in the supplementary material also reveal that InNeRF has a better performance for each scene.

The superior generalization ability of InNeRF is also reflected in qualitative results. As shown in Fig. 2, we compare the performance of methods on rendering the same randomly-selected testing view based on different source-view sets. The results of other approaches contain more obvious artifacts than InNeRF and even become worse in the $S_3$ scenario where the difference between source views and

**Table 1: Quantitative comparison of methods on the scene-agnostic setting for the realistic synthetic dataset [16] and the real forward-facing dataset [15].**

| Dataset | $S_i$ | PSNR ↑ | | | | SSIM ↑ | | | | LPIPS ↓ | | | |
|---|---|---|---|---|---|---|---|---|---|---|---|---|---|
| | | PixelNeRF | MVSNeRF | IBRNet | InNeRF | PixelNeRF | MVSNeRF | IBRNet | InNeRF | PixelNeRF | MVSNeRF | IBRNet | InNeRF |
| realistic synthetic | S1 | 21.20 | 22.47 | 25.31 | **26.45** | 0.857 | 0.874 | 0.913 | **0.922** | 0.161 | 0.143 | 0.104 | **0.092** |
| | S2 | 17.00 | 18.44 | 21.80 | **23.16** | 0.732 | 0.755 | 0.805 | **0.842** | 0.295 | 0.286 | 0.236 | **0.183** |
| | S3 | 15.88 | 17.43 | 20.99 | **22.70** | 0.660 | 0.687 | 0.749 | **0.810** | 0.355 | 0.328 | 0.270 | **0.211** |
| | S4 | 14.67 | 16.25 | 19.97 | **21.72** | 0.567 | 0.597 | 0.672 | **0.758** | 0.440 | 0.400 | 0.322 | **0.248** |
| real forward-facing | S1 | 19.02 | 20.09 | 24.96 | **24.97** | 0.651 | 0.680 | 0.813 | **0.816** | 0.380 | 0.347 | 0.208 | **0.205** |
| | S2 | 16.30 | 17.68 | 22.69 | **22.94** | 0.576 | 0.614 | 0.749 | **0.760** | 0.459 | 0.422 | 0.273 | **0.260** |
| | S3 | 13.56 | 15.21 | 20.33 | **20.81** | 0.489 | 0.543 | 0.683 | **0.701** | 0.551 | 0.504 | 0.340 | **0.318** |

**Table 2: Quantitative comparisons of methods on the per-scene fine-tuning setting for the realistic synthetic dataset [16] and the real forward-facing dataset [15].**

| Dataset | $S_i$ | PSNR ↑ | | | | SSIM ↑ | | | | LPIPS ↓ | | | |
|---|---|---|---|---|---|---|---|---|---|---|---|---|---|
| | | PixelNeRF | MVSNeRF | IBRNet | InNeRF | PixelNeRF | MVSNeRF | IBRNet | InNeRF | PixelNeRF | MVSNeRF | IBRNet | InNeRF |
| realistic synthetic | S1 | 24.06 | 27.04 | 29.27 | **30.79** | 0.877 | 0.913 | 0.940 | **0.952** | 0.140 | 0.103 | 0.076 | **0.064** |
| | S2 | 20.15 | 23.30 | 25.91 | **27.76** | 0.770 | 0.813 | 0.847 | **0.881** | 0.263 | 0.221 | 0.187 | **0.142** |
| | S3 | 19.27 | 22.56 | 25.23 | **27.35** | 0.714 | 0.759 | 0.802 | **0.849** | 0.301 | 0.256 | 0.216 | **0.165** |
| | S4 | 18.23 | 21.57 | 24.33 | **26.65** | 0.639 | 0.689 | 0.739 | **0.803** | 0.358 | 0.306 | 0.254 | **0.195** |
| real forward-facing | S1 | 20.72 | 23.32 | 26.61 | **26.65** | 0.693 | 0.758 | 0.847 | **0.853** | 0.325 | 0.260 | 0.177 | **0.173** |
| | S2 | 18.28 | 21.11 | 24.69 | **24.99** | 0.625 | 0.696 | 0.788 | **0.811** | 0.384 | 0.313 | 0.225 | **0.212** |
| | S3 | 15.66 | 18.62 | 22.62 | **23.25** | 0.544 | 0.623 | 0.727 | **0.767** | 0.458 | 0.377 | 0.276 | **0.256** |

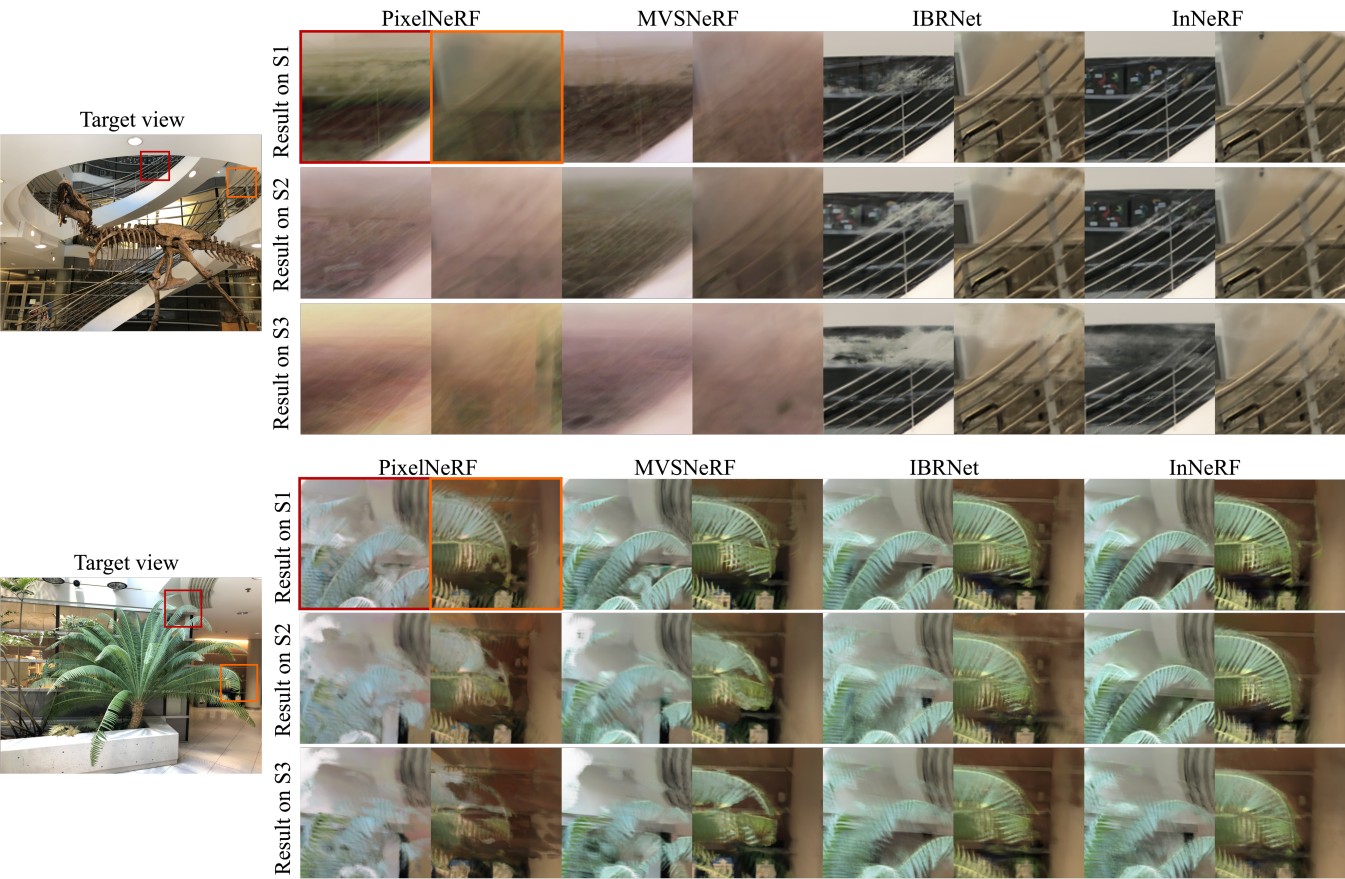

**Figure 2: Qualitative results for the Trex and the Fern scenes [15] under the scene-agnostic setting.**

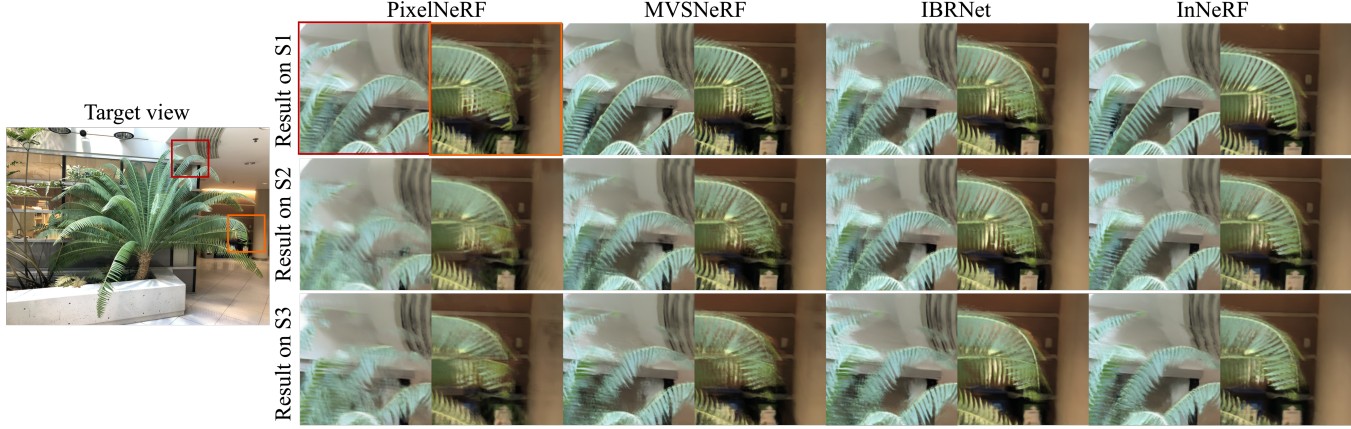

Figure 3: Qualitative results for the Fern scene [15] under the per-scene finetuning setting.

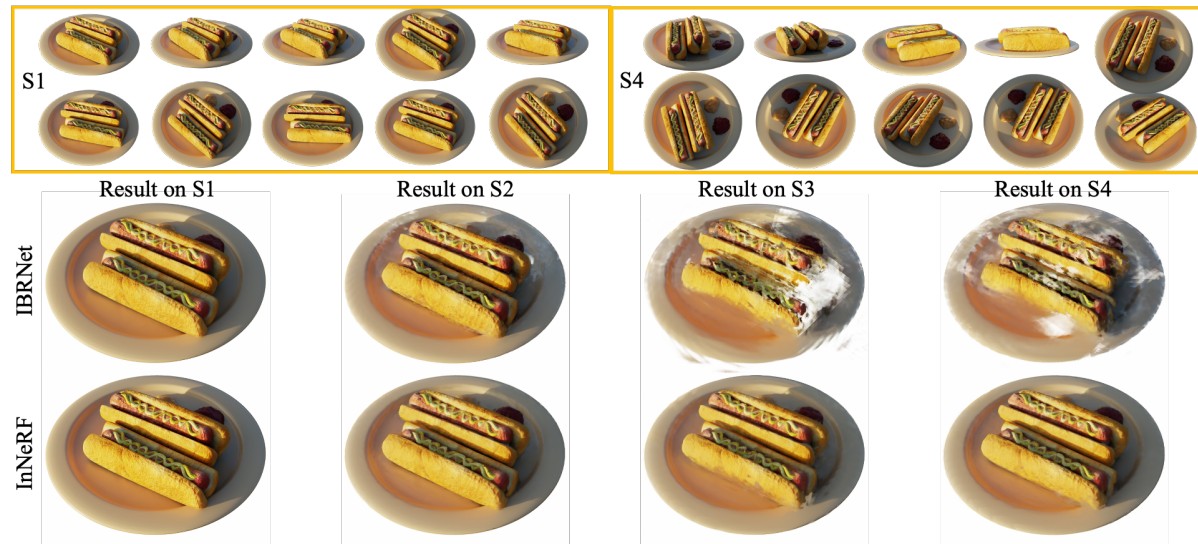

Figure 4: Qualitative results for the Hotdog scene under the per-scene finetuning setting. The source-view sets S1 and S4 are listed in the yellow frame.

the target view is larger than that in $S_1$ and $S_2$. As highlighted in colored frames, other methods cannot synthesize clean boundaries of guardrails and fronds and recover thin structures.

From the above qualitative results, we observe that there exists a gradual degradation in the synthesized view when the difference between source views and the target rendering view increases from $S_1$ to $S_3$. Similarly, in quantitative results from $S_1$ to $S_3$, PSNR and SSIM values both decrease while LPIPS increases for all competing methods. It reveals that the more different the source views are with respect to the target rendering view, the more difficult novel view synthesis becomes. Tab. 1 also indicates that the advantage of InNeRF becomes more significant than other methods with the increase of the difference between source views and the target view. It demonstrates that InNeRF has a strong ability to explore complicated relationships between source views and the target view

and learn a better scene representation in challenging scenarios. More results are provided in the supplementary material.

*4.2.2  Per-scene Finetuning Experiments.* In the per-scene finetuning experiment, pretrained models of competing methods are finetuned for each scene.

As shown in Tab. 2, InNeRF outperforms other methods after per-scene finetuning. Similar to scene-agnostic results, per-scene finetuning results further validate that InNeRF can provide more satisfactory novel view rendering than other methods in different source-view settings. Meanwhile, performance gaps between InNeRF and other methods become larger in contrast with that in the scene-agnostic setting, which indicates that per-scene finetuning can further fulfill the potential of InNeRF. Similar to quantitative results, Fig. 3 shows that InNeRF provides more realistic

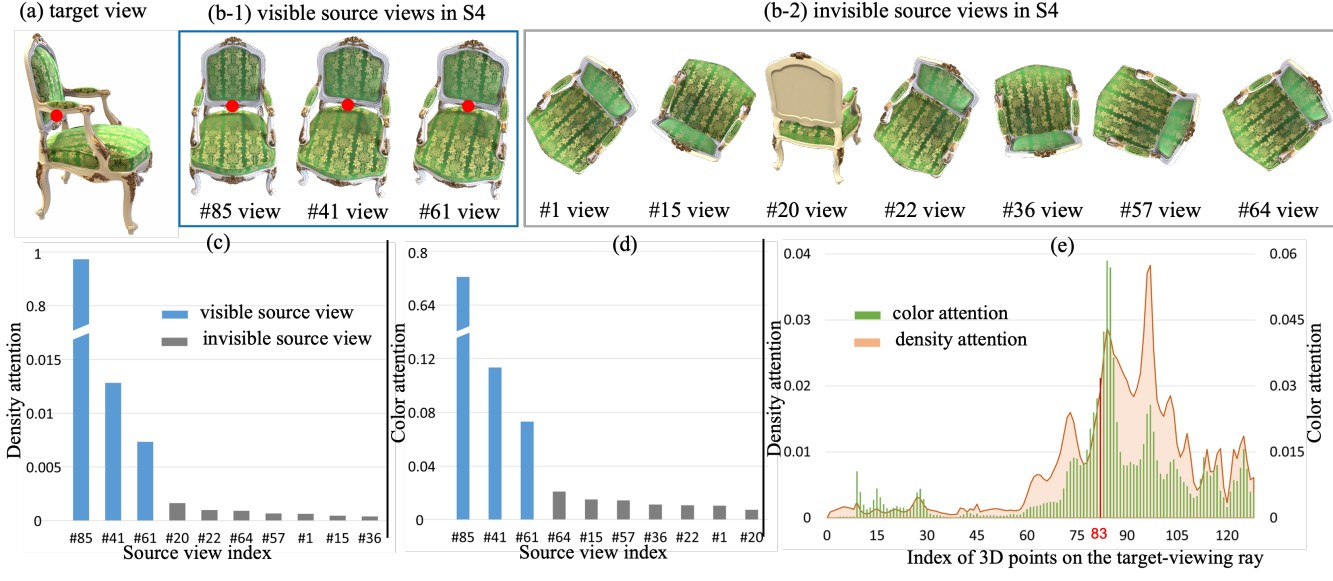

Figure 5: Interpretation results of finetuned InNeRF for a target view of Chair scene based on source-view set S4.

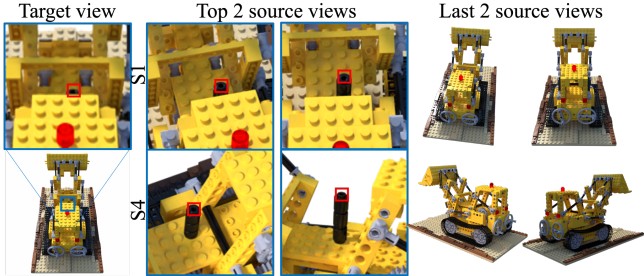

Figure 6: Interpretation results of fine-tuned InNeRF for a target view of the Lego scene.

view synthesis results with fewer artifacts in comparison to other approaches.

In Fig. 4, InNeRF is compared with IBRNet in four source-view sets ($S_1$, $S_2$, $S_3$ and $S_4$). Here, we randomly select one view from the hotdog scene as the target rendering view. To show the difference between source view sets, we display overall source views in $S_1$ and $S_4$ at the bottom of Fig. 4. It is obvious that the view angles of source views in $S_1$ are closer to the rendering view compared with those in $S_4$. In Fig. 4, the top two rows display the rendering views of competing methods based on four source-view sets. The artifacts in the rendering views of IBRNet are perceptible in $S_2$ and become worse in $S_3$ and $S_4$. In contrast, the artifacts in rendering views of InNeRF remain at a low degree in four source-view sets. It illustrates InNeRF can obtain better rendering results than IBRNet in different source-view sets, especially when there is a large difference between the source views and the rendering view.

*4.2.3 Analysis of Interpretability in InNeRF.* Based on the attention mechanism, InNeRF utilizes shape and appearance consistency

in both the surrounding-view space and the ray-cast space, thus improving the model interpretability. Here, we evaluate the interpretability of InNeRF to examine whether it is consistent with human perception.

In the surrounding-view space, we visualize the attention of different source views to a target 3D point to interpret its rendering in $\text{Decoder}_\sigma^{views}$ and $\text{Decoder}_c^{views}$. Similarly, in the ray-cast space, the rendering process of $\text{Decoder}_\sigma^{ray}$ and $\text{Decoder}_c^{ray}$ can be explored by visualizing the attention of surrounding 3D points on the target-viewing ray to the target 3D point. Specifically, for a 2D region (a $5 \times 5$ pixel region) in the rendering view, we first compute the average depth value of the corresponding view directions for the target pixels based on our learned neural radiance field. Then we retrieve the 3D point that is located closest to the average depth in the average viewing direction as the target-interpreted 3D point. For the target 3D point, we can explain its rendering process in both surrounding-view and ray-cast spaces by visualizing the corresponding attention layers in InNeRF.

To analyze the interpretability of InNeRF, we provide interpretation to a randomly selected testing view of the chair scene based on source-view sets $S_4$ in Fig. 5. The target rendering view is shown in Fig. 5 (a) and the target location for interpretation is marked as a red dot. For human visual perception, the source views are divided into two groups depending on whether they capture the target location (red dot) in the rendering view. Fig. 5 (b-1) shows source views that capture the target location, and Fig. 5 (b-2) shows source views that fail to capture.

For the target location (red dot) in Fig. 5 (a), Fig. 5 (c) and (d) display attention of source views to the target location for rendering the query density and color in $\text{Decoder}_\sigma^{views}$ and $\text{Decoder}_c^{views}$, respectively. In Fig. 5 (c) and (d), attention of the visible source views in Fig. 5 (b) are colored blue for clarity. Source views (85, 41, and 61) with high attention values are consistent with those

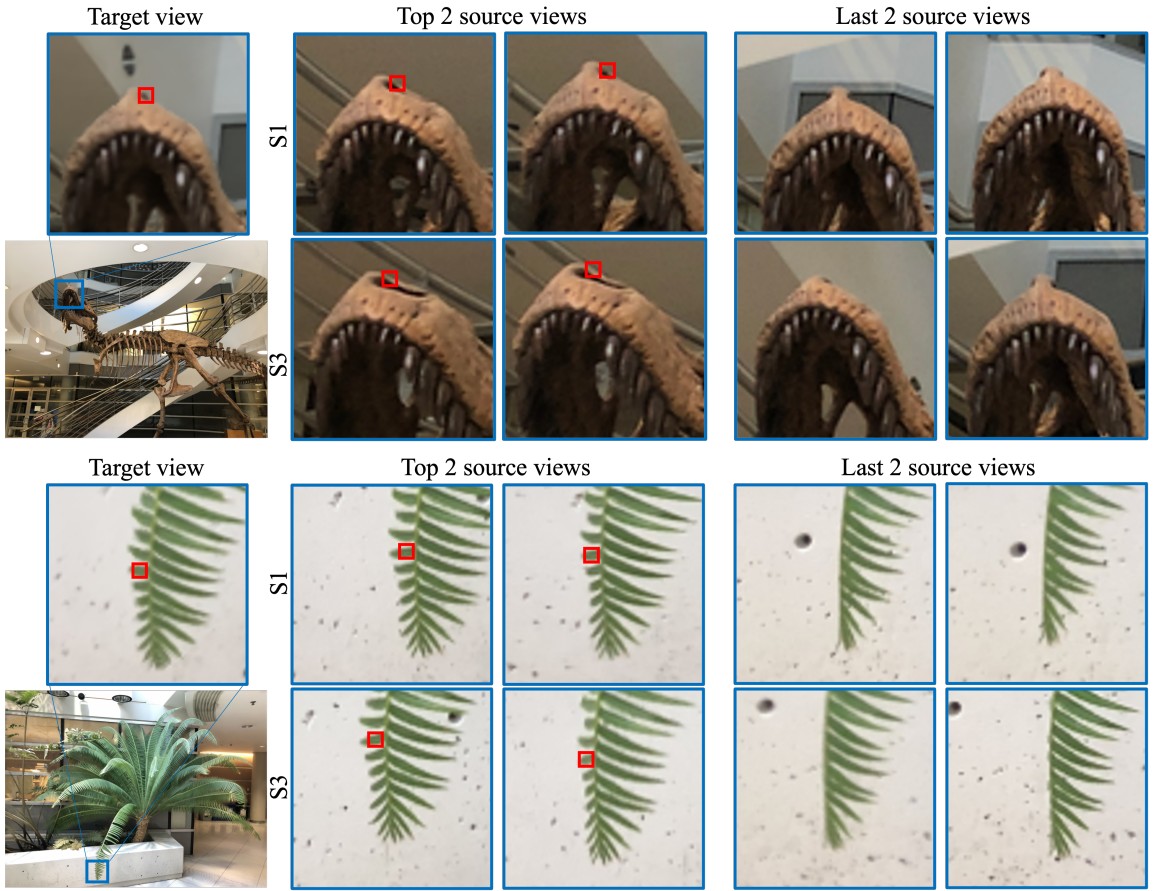

**Figure 7: Interpretation results of InNeRF for (top) the wings of the nose spot, (middle) the leaves on the left side, and (bottom) the black tile in the novel views under the per-scene finetuning setting.**

with the visible target location. It indicates that the attention layers in $\text{Decoder}_\sigma^{views}$ and $\text{Decoder}_c^{views}$ can learn the important source views that meet human perception. Fig. 5 (e) depicts the density attention (green) and color attention (orange) among 3D points along the target-viewing ray for rendering the query 3D point in $\text{Decoder}_\sigma^{ray}$ and $\text{Decoder}_c^{ray}$. Here, the red index (83) denotes the retrieved 3D point for the target location in the rendering view. As shown in Fig. 5 (e), both density attention and color attention in $\text{Decoder}_\sigma^{ray}$ and $\text{Decoder}_c^{ray}$ exist a crest near the query 3D point, which illustrates that InNeRF in the ray-cast space takes into account the consistency of neighbor points when rendering the query point.

Fig. 6 shows the top two source views that are of high-density attention for the target location of Lego in the realistic synthetic dataset, and the last two columns show the last two source views with low-density attention. Given that the top-attention source views capture the target location (red frame), it is reasonable that they receive more attention for the query rendering.

Fig. 7 provides the interpretation results in the forward-facing dataset. The leftmost column shows the rendering view and an enlarged region framed by a blue box. The second and third columns

show the top two source views that are of high-density attention for the target location, and the last two columns show the last two source views with low-density attention. For different source view sets, the top two source views of the framed leaf region both include the corresponding leaf region while the last two source views do not. It indicates that the interpretation results are reasonable for human perception.

## 5 CONCLUSION

We propose a unified Transformer-based NeRF framework to learn a general neural radiance field for novel view synthesis. The proposed framework can explore complex relationships between source views and the target rendering view. Meanwhile, the framework improves intrinsic interpretability by utilizing the shape and appearance consistency of 3D scenes. Experiments demonstrate that InNeRF achieves state-of-the-art performance on real and synthetic datasets in both scene-agnostic and per-scene finetuning settings. In the future, we intend to extend InNeRF to conditional generative radiance fields, employing learned prior knowledge to generate a more expressive and interpretable 3D scene representation for the conditional information.

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
