# OpenReview forum: "InNeRF: Learning Interpretable Radiance Fields for Generalizable 3D Scene Representation and Rendering"
_acmmm.org/ACMMM/2024/Conference — MM2024 Poster_

### Official Review · Reviewer_4BzB · 2024-05-21

**Rating:** 4
**Confidence:** 3

**Summary:**

The paper proposes InNeRF for generalizable novel view synthesis. InNeRF uses an end-to-end interpretable Transformer-based network to unify source-view fusion and target-view rendering. Specifically, it applies attention mechanism in both the surrounding view space and the ray-cast space. For a query 3D point, both its projected 2D
pixels from the surrounding source views and its adjacent 3D points along the query ray are utilized to decode the density and color at this point. Experiments show that InNeRF outperforms prior arts such as PixelNeRF, MVSNeRF and IBRNet.

**Strengths:**

1. The proposed method achieves better performance than PixelNeRF, MVSNeRF and IBRNet, both quantitatively and qualitatively.
2. As demonstrated in Fig. 4, the proposed method produces much better rendering quality than IBRNet when the target view are far from the source views.
3. The paper contains an analysis of interpretability and shows that the learned attention are reasonable.

**Limitations:**

1. As mentioned in Line 326-327, the view-space decoder is designed to be invariant to permutations of source views. However, PE is used in Fig. 1, which will make the output dependent on the order of the source views.
2. It is unclear about the motivation for using ray-cast space decoders. Since 3D points along a ray will be aggregated during volume rendering, why is it necessary to do an extra round of aggregation along the ray before volume rendering? Also, this extra round could increase the time and memory cost.
3. There is a lack of comparisons with more recent methods such as "Generalizable patch-based neural rendering", "Learning to render novel views from wide-baseline stereo pairs" and "Is Attention All That NeRF Needs?".
4. There is no ablation study on the proposed components and the rendering speed is not reported.
5. Missing references to advances in NeRF and few-shot view synthesis:
[1] Zip-nerf: Anti-aliased grid-based neural radiance fields
[2] Neurbf: A neural fields representation with adaptive radial basis functions
[3] Tri-miprf: Tri-mip representation for efficient anti-aliasing neural radiance fields
[4] pixelsplat: 3d gaussian splats from image pairs for scalable generalizable 3d reconstruction
[5] CaesarNeRF: Calibrated Semantic Representation for Few-shot Generalizable Neural Rendering

**Suitability:**

2

---

### Official Review · Reviewer_pwTp · 2024-05-26

**Rating:** 4
**Confidence:** 3

**Summary:**

This paper proposes a Transformer-based NeRF framework called Interpretable Neural Radiance Fields (InNeRF) to learn a general neural field for novel view synthesis. InNeRF investigates the deep relationship between the target-rendering view and source views to enhance the shape and appearance consistency of a 3D scene by the attention mechanism.

**Strengths:**

1. The novelty of this paper is to use the attention mechanism in the Transformer to enhance the relationship of the pixels in the object with the corresponding projections in surrounding views. The implementation combines the Transformer with NeRF and performs the advantage of the attention mechanism well.
2. The demonstration of the workflow is clear and detailed. Equations for different decoders are clear.
3. The explanation for the advantages of InNeRF is clearly demonstrated by the experiment results. For example, the Analysis of Interpretability in InNeRF clearly shows the interpretability ability of InNeRF.
4. There are adequate evaluations for the comparison in the Experiment part.

**Limitations:**

1. The ‘color frames’ mentioned in Lines 623-625 only exist in the results of PixelNeRF. However, there are some blurs for the thin structure of the reconstructed results of MVSNeRF, so there may be some problems in plotting the figure.
2. As mentioned in Lines 856-857, there exists a crest near the query 3D point for both the density decoder and color decoder. From Figure 5(e), the first high and second high wave peaks for the color decoder and density decoder are different. Is there any explanation for this phenomenon?
3. There exists a typo: The ‘Piexls’ of ‘Latent Codes of Projected Piexls at Source Views’ in Module A’s blue square of Figure 1 should be ‘Pixels’.
4. There is no demonstration of the limitations of InNeRF. It would be better if this paper contained this part.

**Suitability:**

3

---

### Official Review · Reviewer_Bc1m · 2024-06-02

**Rating:** 3
**Confidence:** 3

**Summary:**

The paper proposes a novel NeRF-based approach for generalizable scene representation and rendering.

**Strengths:**

- well written
- nice figures

**Limitations:**

- I think the method needs more recent models to compare to. Here is some works to compare proposed method to:
[1] MatchNerf: Explicit Correspondence Matching for Generalizable Neural Radiance Fields
[2] Geometry-aware Reconstruction and Fusion-refined Rendering for Generalizable Neural Radiance Fields

- For example, MatchNeRF also proposes to utilize Transformers. Could you explain more the difference and advantages of your approach?

**Suitability:**

2

---

### Meta-Review · Area_Chair_v35a · 2024-07-04

**Recommendation:** Accept (Poster)
**Confidence:** 5

**Metareview:**

All 3 reviewers suggested acceptance for this paper after rebuttal (2 Weak Accept and 1 Borderline Accept). Congratulations and please revise the paper according to the reviewer's suggestions.